# Temporal trends in aerobic physical activity guideline adherence among nationally representative samples of U.S adults between 2011 and 2019: Cross-sectional findings from a sample of over 2 million adults

**David Abernethy**[1]*, **Jason Bennie**[2], **Toby Pavey**[3]

1 School of Exercise and Nutrition Science, Queensland University of Technology, Brisbane, Australia, 2 Murrumbidgee Primary Health Network, Wagga Wagga, Australia, 3 Faculty of Health, Queensland University of Technology, Brisbane, Australia

* David.abernethy@hdr.qut.edu.au

**Data Availability Statement:** All BRFSS data files are publicly available from the BRFSS index webpage https://www.cdc.gov/brfss/index.html.

## Abstract

### Background

Physical inactivity is a significant public health concern associated with numerous adverse health outcomes and substantial economic costs. This study describes the prevalence, trends and correlates for adherence to moderate to vigorous physical activity (MVPA) guidelines among a large sample of U.S. adults.

### Methods

Data from the 2011, 2013, 2015, 2017, and 2019 Behavioral Risk Factor Surveillance System surveys were analyzed. Self-reported MVPA was assessed by the same item across each survey. Population-weighted prevalence was calculated for meeting MVPA guidelines (150+ mins/wk). Adjusted prevalence ratios for reporting sufficient MVPA across sociodemographic, behavioral and health variables were calculated by multivariate Poisson regression.

### Results

Data was available for 2,052,288 respondents (≥ 18 years). Across the surveys, the prevalence of sufficient MVPA fluctuated but remained between 49.5% and 51.1%. Among those aged 18 to 24, the prevalence of sufficient MVPA declined between surveys, from 56.5% in 2011 to 49.7% in 2019. Notable correlates of adhering to guidelines included male sex, higher education, former and never smokers, normal body mass index and increased fruit and vegetable consumption.

**Funding:** The author(s) received no specific funding for this work.

**Competing interests:** The authors have declared that no competing interests exist.

## Conclusion

From 2011 to 2019, approximately half of US adults reported sufficient MVPA, with a steady decline observed among young adults. While many identified correlates of adhering to PA guidelines were observed, this study has provided further evidence for correlates that had previously provided inconsistent or inconclusive results. These findings emphasize the complexity of addressing physical inactivity and the importance of multifaceted public health strategies tailored to diverse populations.

## Introduction

The public health burden of physical inactivity has become increasingly apparent, with approximately 6–10% of all coronary heart disease, type two diabetes and breast cancer cases attributable to this modifiable risk factor alone [1]. In addition to contributing to more than three million deaths and 32 million disability-adjusted life years every year, physical inactivity has been identified as the fourth leading risk factor for mortality [2]. Moreover, the economic and healthcare-related costs associated with inactivity are high. For example, recent projections suggest that almost 500 million new cases of noncommunicable diseases will occur worldwide by 2030, accompanied by direct health-care costs of $520 billion dollars if the prevalence of inactivity remains the same [3].

Epidemiological evidence has indicated the protective benefits of physical activity (PA) against chronic disease, regardless of intensity, frequency and dosage [4–6], with risk reductions of between 20%-30% for at least twenty-five chronic conditions associated with regular engagement in PA [7]. A similar relationship exists for mortality, with a dose-response relationship between cardiovascular- and all-cause mortality and PA [6, 7]. Improvements in brain, muscle, bone and mental health have also been attributed to regular engagement in PA [4]. While at present, there is no exact dose of PA for optimal health benefits and longevity, the recently updated 2020 World Health Organization (WHO) guidelines on PA and sedentary behavior for adults aged 18 to 64 recommend adults engage in at least 150 to 300 minutes of moderate-intensity or 75 to 150 minutes of vigorous-intensity aerobic PA or a combination of the two and incorporate muscle-strengthening activities on two or more days per week [8]. These guidelines were developed by synthesizing high-quality evidence from longitudinal cohort studies, that have been shown to reduce the risk of adverse health outcomes and increase longevity [8].

Although many studies have identified the association between regular PA and health benefits, others have investigated how increasing PA doses from currently low or no activity can largely reduce risks of adverse health outcomes. One systematic review concluded that clinically relevant health benefits could be experienced from half or less than half of current MVPA recommendations, with the largest relative health benefits experienced at the lowest doses of MVPA [6]. While limited evidence exists suggesting a U-shaped association between PA and the risk of adverse events [9], the evidence above highlights the benefits of PA, regardless of intensity, frequency and dosage, and signals the importance of regular engagement in PA behaviors.

Despite the benefits of PA being well known, experts have stated that the issue of physical inactivity has reached pandemic proportions [10]. Public health estimates have shown that adults are not engaging in sufficient PA, with little improvement made within the last decade

and insufficient PA trends persisting [11]. In response to global concerns regarding insufficient levels of activity, the WHO developed a global action plan promoting PA. The primary aim of this plan is to increase activity levels by 10 and 15% by 2025 and 2030, respectively [8]. However, experts have stated that goals and targets set by governments and international agencies are more prone to be amended than accomplished [12]. While the prevalence of inactivity varies considerably between countries and geographical regions, recent global estimates suggest that 27.5% of adults worldwide are not meeting the recommended dose of PA to improve health and protect against adverse effects [8]. The prevalence of insufficient activity is among the greatest in the United States, with current adult adherence to PA guidelines estimates based on self-report ranging from 50.5% [13] to 65.2% [14].

The current state of evidence about correlates of aerobic PA in adults has underscored the multifaceted nature of factors that influence consistent engagement in PA, as many studies have highlighted the significance of sociodemographic factors (age, sex, level of education, poor health status), behavioral and psychosocial factors (self-efficacy, perceived behavioral control, social support) [7, 15]. While many correlates of activity have been repeatedly identified in various studies, others have provided results that were conflicting or inconclusive. For example, many individual-level factors such as sex, level of education and health status have been frequently assessed, with increasing male sex, increased education and poorer health status repeatedly identified as having an association with PA [15, 16]. While age is largely considered to be inversely associated with PA [15, 17, 18], limited evidence has suggested that no association exists between the variables [19]. Some behavioral and health variables have also provided mixed results about their association with PA. Smoking status has been included as a potential correlate in a range of studies investigating potential associations with PA, however, studies have continually concluded the available evidence is limited and/or inconclusive [16, 20]. Inconclusive or inconsistent research findings in this context can contribute to difficulties in determining areas for public health intervention and which factors are most influential in promoting PA.

Ongoing research of PA trends in populations and identifying correlates of sufficient activity help to refine the understanding of shifts in behavior, priority areas and populations requiring intervention and the impact of public health campaigns targeting physical inactivity. Therefore, this study aims to (i) describe the proportion of US adults adhering to current aerobic PA guidelines from 2011 to 2019; and (ii) identify significant sociodemographic, behavioral, and health-related correlates for guideline adherence in a representative sample of two million adults living in the United States.

## Methods

The Behavioral Risk Factor Surveillance System BRFSS is an annual cross-sectional public health survey conducted in the United States that collects data from residents about health-related risk behaviors, chronic health conditions and the use of preventive health services [21]. Developed in 1984, the BRFSS is the largest continuously conducted health survey of its kind in the world. For the present analysis, data from the 2011, 2013, 2015, 2017 and 2019 individual surveys were analyzed. This is due to the BRFSS protocol, which collects comprehensive PA data biennially. Each BRFSS survey is approved by the National Center for Health Statistics Research Ethics Review Board [21]. The response rates were 49.7%, 45.9%, 47.2%, 45.9%, and 49.4% for the 2011, 2013, 2015, 2017 and 2019 BRFSS surveys, respectively [21].

The same survey and methodology were used across all the BRFSS surveys used in this research to assess self-reported data for PA, height and weight, sociodemographic and lifestyle variables [21]. Due to the BRFSS's large scope and depth of questioning, it is commonly used

in health research to explore relationships between various exposures and their impact on health [21]. This survey has been repeatedly assessed to ensure that its items are valid and reliable, which produces data that allows for study results to be replicated and accurate. Several studies have previously shown that the assessment items used in the BRFSS display evidence of moderate or high levels of reliability and validity [22–24].

Data were available for 2,307,980 respondents; however, for the analyses performed in this study, participants were excluded if they were missing data for PA (n = 255,692) (final sample = 2,052,288). Individuals missing data for the explanatory variables were excluded during the analysis. To enhance the generalizability of this research, no other inclusion or exclusion criteria were applied to the initial sample.

## Physical activity assessments

Moderate to vigorous physical activity (MVPA) of BRFSS respondents was assessed using previously validated questionnaires [25]. The survey questions have acceptable test-retest reliability and concurrent validity for classifying groups to levels of recommended MVPA when compared to accelerometry as the gold standard [25].

When examining an individual's PA habits, the BRFSS surveyors provided participants with the following statement: "*The next few questions are about exercise, recreation or physical activities other than your regular job duties.*" This was followed by "*During the past month, other than your regular job, did you participate in any physical activities or exercises such as running, calisthenics, golf, gardening or walking for exercise?*". If respondents replied yes, further questions were asked: "*What type of physical activity or exercise did you spend the most time doing during the past month?*", "*How many times per week or per month did you take part in this activity during the past month?*", and "*When you took part in this activity, for how many minutes or hours did you usually keep at it?*"

The same set of questions was asked about a second activity when respondents indicated that they engaged in more than one PA. All reported time spent being physically active was reported in hours and minutes, with all activities coded as being either "aerobic" or "nonaerobic" according to a list of 56 activities [26, 27]. Aerobic activities included exercises such as walking, running, swimming, and cycling, whereas nonaerobic activities included activities such as bowling, golf, house chores and weight training. MVPA included only aerobic activities and was classified as moderate or vigorous based on the estimated metabolic equivalent (MET) [21]. Activities classified as moderate intensity had a MET value greater than 3.0 but lower than the cutoff for vigorous activities. Activities classified as vigorous intensity had an allocated MET value of 6.0 or higher, to reflect at least 60% of an individual's maximal cardiorespiratory capacity, based on sex and age [27]. Total MVPA was calculated by multiplying weekly minutes of vigorous-intensity activity by two and then adding to moderate-intensity weekly minutes, to reflect the higher MET from vigorous activities.

## Physical activity categories

Two mutually exclusive PA categories were formed according to participant adherence to the World Health Organization's aerobic MVPA guidelines. Individuals who failed to record 150 or more minutes of MVPA per week were categorized as "not meeting guidelines" (MVPA = 0.00–149.99 min/wk), and individuals who recorded 150 minutes or more were categorized as "meeting guidelines" (MVPA = 150 $\geq$ min/wk).

## Explanatory variables

Sociodemographic, lifestyle and health characteristics were assessed using standardized survey items. All of the explanatory variables included have been recognized as having an association with PA behaviors [15]. Sociodemographic variables included age (18–24, 25–34, 35–44, 45–54, 55–64, 65 or older), sex (male, female), level of education (did not graduate high school, graduated high school, attended technical school or college, graduated technical school or college), race (white, black, multiracial, Hispanic, other), employment status (employed, unemployed, homemaker, student, retired, unable to work), marital status (married/member of an unmarried couple, divorced/widowed/separated, never married), and number of children (zero, one, two, three or more). Lifestyle variables included smoking status (current smokers, former smokers and never smoked) and fruit and vegetable consumption. The BRFSS measures daily fruit and vegetable consumption with six items assessing the frequency of consuming 100% fruit juice, fruit, beans (legumes), dark green vegetables, orange vegetables and other vegetables for the month before respondent interviews [28]. The daily number of fruit and vegetable servings was created by adding the total calculated number of daily fruit servings to the total calculated number of vegetable servings. Health-related variables included self-rated health (SRH) (excellent, very good, good, fair, poor), number of poorer mental health days, number of chronic health conditions and body mass index (BMI). Days with poor mental health were assessed by asking respondents "*now thinking about your mental health, which includes stress, depression, and problems with emotions, for how many days during the past 30 days was your mental health not good?*" (zero, one to two, three to six, seven to fourteen, fifteen to thirty) [29]. The twelve chronic health conditions assessed include hypertension, high cholesterol, heart attack, coronary heart disease, stroke, asthma, cancer (non-skin), chronic obstructive pulmonary disorder, depression, kidney disease, diabetes and arthritis. These conditions were included due to their recognized association with morbidity and mortality [30] and assessed by asking respondents "*Has a doctor, nurse or other health professionals ever told you that you had any of the following?*" [21]. BMI was calculated from self-reported height and weight data using the standardized formula ($kg/m^2$).

## Statistical analysis

All statistical analyses were completed using IBM's SPSS (version 29). Each participant was given an individual weighting factor to be used for analyses, effected to help correct for non-response, stratification, and clustering, which increased population representativeness and reduced the risk of bias. More detailed information concerning the weighting of BRFSS respondents is available elsewhere [31]. Weighted prevalence levels and 95% confidence intervals for adhering to MVPA guidelines in each BRFSS survey were reported for the whole sample and all subcategories of the various explanatory variables included.

Multivariate Poisson log-linear regression analyses with a robust-error variance were used to calculate adjusted prevalence ratios (APRs) for meeting the PA guidelines for each category of the explanatory variables compared to reference categories, while also adjusting for all other explanatory variables. The use of prevalence ratios obtained from Poisson regression has been identified as a more statistically robust method in cross-sectional epidemiology studies compared to general logistic regression [32]. P-values were based on two-sided tests and were considered statistically significant at p < 0.05. Before performing analyses, collinearity between the various explanatory variables was examined using variance inflation factor (VIF), with VIFs $\geq$ 2.5 indicating multicollinearity [33]. VIFs ranged from 1.05–2.14, demonstrating no evidence of collinearity.

## Results

### Characteristics of the sample

Data were available for 2,052,288 individuals ($\geq$ 18 years old) and are shown in Table 1. In summary, 19.6% were aged 65 years or older, 51.4% were female, and most identified as either white, black, or Hispanic (92.2%). Over half of the sample reported never having smoked (58.3%), and 18.3% self-reported having excellent health.

### Intensity of reported PA

Across each survey, more than 60% of respondents engaged in only moderate PA (range = 62.6%– 63.3%), while approximately 17.5% (range = 16.9% - 18.6%) engaged in only vigorous activities. The remaining 19.5% of respondents (range = 18.1% - 20.5) engaged in a combination of moderate and vigorous activities.

### Most common activities

The two most common aerobic activities for BRFSS respondents for all surveys were walking (range = 60.5% - 63.3%) and running (range = 15.4% - 16.1%). Cycling on a bicycle or machine (range = 8.5% - 9.6%) and gardening (range = 7.7% - 10.7%) were the next most reported activities by respondents.

### Adherence to aerobic PA guidelines

Weight-adjusted adherence (95% CI) to the aerobic PA guidelines for each of the explanatory variables is available in Table 2. Guideline adherence for the total sample was 51.1% (50.8–51.4), 49.5% (49.2–49.8), 50.6% (50.2–50.9), 49.7% (49.3–50.0) and 50.1% (49.7–50.4) for the 2011, 2013, 2015, 2017 and 2019 BRFSS surveys. The prevalence of reporting sufficient PA declined for respondents aged 18 to 24 across the surveys, dropping from 56.5% (55.3–57.6) in 2011, to 49.7% (48.6–50.8) in 2019. Declines in sufficient PA were also observed across the surveys for single individuals and those with excellent self-reported health, dropping from 52.5% (51.7–53.3) to 47.6% (46.9–48.3) and 63.1% (62.4–63.8) to 60.9% (60.1–61.7), respectively. The prevalence of reporting sufficient PA increased for respondents aged 65 and above across the surveys, climbing from 51.9% (51.4–52.4) in 2011, to 53.9% (53.4–54.5) in 2019. Retired individuals also reported increased guideline adherence from 2011 to 2019, increasing from 55.4% (54.9–55.9) to 56.6% (55.9–57.1).

### Correlates of physical activity guideline adherence

Adjusted prevalence ratios (APRs) and 95% confidence intervals for meeting the MVPA guidelines by categories of each individual explanatory variable are presented in Table 3.

### Sociodemographic variables

The prevalence of MVPA guideline adherence was similar for all age groups across each of the surveys included, with individuals aged between 25–34 and 35–44 slightly less likely to meet guidelines. In the last data collection (2019), respondents aged 55 or older reported a notably higher prevalence of engaging in 150 or more minutes of MVPA compared to the 18-24-year-old age category (APR $\geq$ 1.09). Throughout all surveys, women were less likely to meet guidelines than men (APRs $\leq$ 0.95). Compared to those identifying as white, those in all other race/ethnicity categories were less likely to meet PA guidelines (APRs $\leq$ 0.95), aside from those who were multiracial (APRs $\geq$ 1.07).

**Table 1. BRFSS sample characteristics by survey year (2011, 2013, 2015, 2017 and 2019).**

| Variable | Survey Year | | | | |
|---|---|---|---|---|---|
| | **2011** | **2013** | **2015** | **2017** | **2019** |
| | %[a] (95% CI) | %[a] (95% CI) | %[a] (95% CI) | %[a] (95% CI) | %[a] (95% CI) |
| **Physical activity** | | | | | |
| Did not meet aerobic guidelines | 48.9 (48.6–49.2) | 50.5 (50.2–50.8) | 49.6 (49.3–49.9) | 50.3 (50.0–50.7) | 49.9 (49.6–50.3) |
| Met aerobic guidelines | 51.1 (50.8–51.4) | 49.5 (49.2–49.8) | 50.4 (50.1–50.7) | 49.7 (49.3–50.0) | 50.1 (49.7–50.4) |
| **Sociodemographic variables** | | | | | |
| Age, years | | | | | |
| 18–24 | 12.9 (12.6–13.1) | 13.0 (12.8–13.3) | 12.8 (12.6–13.0) | 12.5 (12.3–12.8) | 12.2 (12.0–12.5) |
| 25–34 | 17.6 (17.4–17.9) | 17.2 (17.0–17.5) | 17.3 (17.1–17.6) | 17.3 (17.1–17.6) | 17.4 (17.2–17.7) |
| 35–44 | 17.6 (17.3–17.8) | 16.6 (16.4–16.9) | 16.4 (16.2–16.6) | 16.3 (16.1–16.6) | 16.3 (16.1–16.5) |
| 45–54 | 18.9 (18.7–19.1) | 18.0 (17.8–18.3) | 17.3 (17.1–17.5) | 16.6 (16.4–16.9) | 16.1 (15.9–16.3) |
| 55–64 | 15.5 (15.3–15.6) | 16.5 (16.3–16.7) | 16.6 (16.4–16.8) | 16.7 (16.5–16.9) | 16.5 (16.3–16.7) |
| ≥ 65 | 17.6 (17.4–17.7) | 18.6 (18.4–18.8) | 19.6 (19.4–19.8) | 20.5 (20.3–20.7) | 21.4 (21.2–21.6) |
| Sex | | | | | |
| Male | 48.7 (48.4–49.0) | 48.6 (48.3–48.9) | 48.7 (48.4–49.0) | 48.7 (48.3–49.0) | 48.7 (48.4–49.0) |
| Female | 51.3 (51.0–51.6) | 51.4 (51.1–51.7) | 51.3 (51.0–51.6) | 51.3 (51.0–51.7) | 51.3 (51.0–51.6) |
| Race | | | | | |
| White | 66.2 (65.9–66.5) | 64.3 (64.0–64.6) | 63.6 (63.3–63.9) | 62.6 (62.3–62.9) | 61.9 (61.5–62.2) |
| Black | 11.3 (11.1–11.5) | 11.6 (11.4–11.8) | 11.7 (11.5–12.0) | 11.8 (11.6–12.0) | 11.8 (11.6–12.0) |
| Other | 5.7 (5.5–5.9) | 6.2 (6.1–6.4) | 6.6 (6.4–6.8) | 7.1 (6.9–7.3) | 7.2 (7.0–7.4) |
| Multiracial | 1.5 (1.4–1.6) | 1.4 (1.3–1.4) | 1.4 (1.4–1.5) | 1.4 (1.4–1.5) | 1.3 (1.3–1.4) |
| Hispanic | 15.3 (15.1–15.6) | 16.5 (16.3–16.8) | 16.6 (16.4–16.9) | 17.0 (16.8–17.3) | 17.8 (17.5–18.0) |
| Employment | | | | | |
| Employed | 55.1 (54.8–55.4) | 55.9 (55.6–56.2) | 56.6 (56.3–56.9) | 56.9 (56.6–57.2) | 57.6 (57.3–57.9) |
| Unemployed | 9.2 (9.0–9.4) | 7.5 (7.4–7.7) | 5.9 (5.8–6.1) | 5.7 (5.5–5.8) | 5.1 (5.0–5.3) |
| Student | 5.8 (5.7–6.0) | 5.9 (5.7–6.1) | 5.8 (5.6–5.9) | 5.8 (5.7–6.0) | 5.4 (5.3–5.6) |
| Homemaker | 7.0 (6.8–7.1) | 6.9 (6.7–7.0) | 6.8 (6.7–7.0) | 6.4 (6.3–6.6) | 5.8 (5.7–6.0) |
| Retired | 16.4 (16.2–16.6) | 16.9 (16.8–17.1) | 17.9 (17.7–18.1) | 18.2 (18.0–18.4) | 19.1 (18.9–19.3) |
| Unable to work | 6.6 (6.4–6.7) | 6.9 (6.7–7.0) | 6.9 (6.8–7.1) | 7.0 (6.8–7.1) | 6.9 (6.8–7.0) |
| Education | | | | | |
| Did not graduate high school | 15.4 (15.2–15.7) | 15.2 (14.9–15.4) | 14.4 (14.1–14.6) | 13.6 (13.3–13.9) | 12.9 (12.7–13.2) |
| Graduated high school | 29.2 (29.0–29.5) | 28.5 (28.3–28.8) | 28.3 (28.0–28.6) | 28.0 (27.7–28.3) | 27.8 (27.5–28.1) |
| Attended college or technical school | 30.0 (29.7–30.2) | 30.8 (30.5–31.1) | 31.0 (30.7–31.3) | 31.0 (30.7–31.3) | 30.9 (30.6–31.2) |
| Graduated college or technical school | 25.4 (25.2–25.6) | 25.5 (25.3–25.7) | 26.3 (26.1–26.6) | 27.3 (27.1–27.6) | 28.4 (28.1–28.6) |
| Marital status | | | | | |
| Married/partnered | 55.4 (55.1–55.7) | 56.0 (55.7–56.3) | 55.6 (55.3–55.9) | 55.3 (55.0–55.7) | 55.3 (55.0–55.6) |
| Widowed/separated/divorced | 19.9 (19.7–20.1) | 20.4 (20.2–20.6) | 20.4 (20.1–20.6) | 20.3 (20.1–20.6) | 20.1 (19.9–20.3) |
| Single | 24.7 (24.4–25.0) | 23.6 (23.3–23.9) | 24.1 (23.8–24.3) | 24.3 (24.0–24.6) | 24.6 (24.3–24.9) |
| Number of children | | | | | |
| Zero | 61.5 (61.2–61.8) | 62.6 (62.3–62.8) | 63.2 (62.9–63.5) | 63.6 (63.2–63.9) | 64.3 (64.0–64.6) |
| One | 15.7 (15.5–15.9) | 15.5 (15.3–15.7) | 15.2 (14.9–15.4) | 15.1 (14.8–15.3) | 14.5 (14.3–14.8) |
| Two | 13.9 (13.7–14.1) | 13.3 (13.1–13.5) | 13.0 (12.8–13.2) | 12.7 (12.5–12.9) | 12.5 (12.3–12.8) |
| Three or more | 8.8 (8.6–9.0) | 8.7 (8.5–8.8) | 8.6 (8.4–8.8) | 8.6 (8.5–8.8) | 8.7 (8.5–8.8) |
| **Behavioral variables** | | | | | |
| Smoking status | | | | | |
| Current smoker | 20.1 (19.8–20.3) | 18.1 (17.9–18.4) | 16.7 (16.5–17.0) | 16.3 (16.1–16.6) | 15.3 (15.0–15.5) |
| Former smoker | 24.7 (24.5–24.9) | 24.7 (24.5–25.0) | 24.6 (24.3–24.8) | 23.9 (23.6–24.2) | 24.2 (23.9–24.4) |

(*Continued*)

**Table 1.** (Continued)

| Variable | Survey Year | | | | |
|---|---|---|---|---|---|
| | **2011** | **2013** | **2015** | **2017** | **2019** |
| | %ᵃ (95% CI) | %ᵃ (95% CI) | %ᵃ (95% CI) | %ᵃ (95% CI) | %ᵃ (95% CI) |
| Never smoked | 55.2 (54.9–55.5) | 57.1 (56.8–57.4) | 58.7 (58.4–59.0) | 59.8 (59.5–60.1) | 60.5 (60.2–60.8) |
| Fruit and vegetable consumption | | | | | |
| Zero | 0.6 (0.6–0.7) | 0.6 (0.6–0.7) | 0.6 (0.6–0.7) | 0.7 (0.7–0.8) | 0.9 (0.8–0.9) |
| One | 30.3 (30.0–30.6) | 30.0 (29.7–30.2) | 30.5 (30.2–30.8) | 25.3 (25.0–25.6) | 27.2 (26.9–27.5) |
| Two | 42.4 (42.1–42.7) | 43.9 (43.6–44.2) | 43.5 (43.2–43.8) | 40.7 (40.4–41.0) | 41.4 (41.0–41.7) |
| Three or more | 26.7 (26.4–27.0) | 25.6 (25.3–25.9) | 25.3 (25.0–25.6) | 33.3 (32.9–33.6) | 30.6 (30.3–30.9) |
| **Health variables** | | | | | |
| Self-rated health | | | | | |
| Excellent | 18.9 (18.7–19.2) | 18.7 (18.5–19.0) | 18.7 (18.5–19.0) | 17.9 (17.7–18.2) | 17.4 (17.1–17.6) |
| Very good | 31.5 (31.2–31.7) | 31.8 (31.6–32.1) | 31.9 (31.6–32.1) | 31.3 (31.0–31.6) | 31.5 (31.2–31.7) |
| Good | 31.4 (31.2–31.7) | 31.3 (31.0–31.5) | 31.7 (31.4–32.0) | 32.1 (31.8–32.4) | 32.3 (32.1–32.6) |
| Fair | 13.3 (13.1–13.5) | 13.4 (13.2–13.6) | 13.1 (12.9–13.3) | 13.8 (13.6–14.0) | 14.1 (13,9–14.3) |
| Poor | 4.9 (4.8–5.0) | 4.8 (4.7–4.9) | 4.6 (4.5–4.7) | 4.9 (4.7–5.0) | 4.7 (4.6–4.8) |
| Days of poorer mental health | | | | | |
| Zero | 64.3 (64.0–64.6) | 66.3 (66.0–66.5) | 65.8 (65.6–66.1) | 64.5 (64.2–64.9) | 61.8 (61.5–62.1) |
| One to two | 8.9 (8.7–9.1) | 8.4 (8.2–8.5) | 8.5 (8.3–8.7) | 8.4 (8.2–8.6) | 8.6 (8.4–8.7) |
| Three to six | 9.6 (9.4–9.8) | 8.9 (8.7–9.1) | 9.1 (9.0–9.3) | 9.2 (9.0–9.4) | 10.0 (9.8–10.2) |
| Seven to fourteen | 5.9 (5.7–6.0) | 5.6 (5.5–5.8) | 5.7 (5.6–5.9) | 6.2 (6.0–6.3) | 6.7 (6.5–6.9) |
| Fifteen to thirty | 11.3 (11.1–11.5) | 10.8 (10.7–11.0) | 10.8 (10.6–11.0) | 11.7 (11.5–12.0) | 12.9 (12.7–13.1) |
| Body mass index | | | | | |
| Normal weight | 35.5 (35.2–35.8) | 34.9 (34.6–35.2) | 34.3 (34.0–34.5) | 33.3 (33.0–33.6) | 32.1 (31.8–32.4) |
| Overweight | 36.5 (36.2–36.8) | 36.3 (36.0–36.5) | 36.4 (36.1–36.7) | 36.0 (35.7–36.3) | 36.0 (35.6–36.3) |
| Obesity class I | 17.4 (17.2–17.7) | 17.7 (17.5–18.0) | 18.1 (17.8–18.3) | 18.3 (18.1–18.6) | 19.0 (18.7–19.2) |
| Obesity class II | 6.4 (6.3–6.6) | 6.8 (6.6–6.9) | 6.8 (6.6–6.9) | 7.6 (7.4–7.8) | 7.8 (7.6–7.9) |
| Obesity class III | 4.1 (4.0–4.2) | 4.3 (4.2–4.5) | 4.5 (4.4–4.7) | 4.8 (4.6–4.9) | 5.3 (5.1–5.4) |
| Number of chronic conditions | | | | | |
| Zero | 35.9 (35.6–36.2) | 34.6 (34.3–34.9) | 0.6 (0.6–0.7) | 34.7 (34.4–35.0) | 33.9 (33.6–34.2) |
| One | 24.6 (24.3–24.8) | 24.6 (24.3–24.8) | 30.5 (30.2–30.8) | 24.3 (24.1–24.6) | 24.7 (24.4–25.0) |
| Two | 16.2 (16.0–16.4) | 16.6 (16.3–16.8) | 43.5 (43.2–43.8) | 16.6 (16.4–16.8) | 16.9 (16.7–17.2) |
| Three or more | 23.3 (23.1–23.6) | 24.3 (24.0–24.5) | 25.3 (25.0–25.6) | 24.4 (24.1–24.6) | 24.4 (24.2–24.7) |

ᵃ Data weighted using stratum weight provided by the Centers for Disease Control and Prevention (CDC)

ᵇ Missing cases as follows: Sex = 63 (0.0%), Race = 28,599 (1.4%), Employment = 9,619 (0.5%), Education = 4,509 (0.2%), Marital Status = 8,693 (0.4%), Number of Children = 7,348 (0.4%), Smoking Status = 10,195 (3.2%), Fruit and Vegetable Consumption = 58,938 (2.9%), Self-Rated Health = 5,775 (0.3%), Poorer Mental Health Days = 32,581 (1.6%), Body Mass Index = 138,556 (6.8)

Students, retirees, homemakers, and unemployed individuals (APR range = 1.05–1.16) were more likely to adhere to PA guidelines than those who were employed for wages (reference category), or unable to work (APRs ≥ 0.93). Respondents who graduated college or technical school were most likely to meet guidelines (APRs ≥ 1.24). Sufficient MVPA did not notably differ between individuals who were single, married/partnered or widowed/separated/divorced (APR range = 0.95–1.02). Aside from 2017 (APR range = 0.96–0.98), the number of children under the age of eighteen living in a household showed to have little effect on the prevalence of adhering to MVPA guidelines for each survey (APR range = 0.98–1.01).

**Table 2. Weighted proportions for reporting sufficient physical activity among BRFSS samples: By sociodemographic, behavioral and health characteristics.**

| | Sufficient MVPA (≥150 minutes/wk.) | | | | |
|---|---|---|---|---|---|
| | **2011** | **2013** | **2015** | **2017** | **2019** |
| | **% (95%CI)[a]** | **% (95%CI)[a]** | **% (95%CI)[a]** | **% (95%CI)[a]** | **% (95%CI)[a]** |
| **Total** | 51.1 (50.8–51.4) | 49.5 (49.2–49.8) | 50.6 (50.2–50.9) | 49.7 (49.3–50.0) | 50.1 (49.7–50.4) |
| *Sociodemographic variables* | | | | | |
| Sex | | | | | |
| Male | 52.6 (52.2–53.1) | 51.0 (50.6–51.5) | 51.4 (50.9–51.8) | 51.0 (50.5–51.5) | 51.9 (51.4–52.4) |
| Female | 49.6 (49.2–50.0) | 48.1 (47.7–48.5) | 49.5 (49.1–49.9) | 48.4 (47.9–48.9) | 48.3 (47.8–48.8) |
| Age, years | | | | | |
| 18–24 | 56.5 (55.3–57.6) | 53.4 (52.3–54.5) | 53.0 (51.9–54.1) | 52.3 (51.1–53.4) | 49.7 (48.6–50.8) |
| 25–34 | 49.3 (48.5–50.2) | 47.9 (47.1–48.7) | 49.0 (48.1–49.8) | 48.0 (47.1–48.9) | 47.2 (46.3–48.0) |
| 35–44 | 49.3 (48.6–50.1) | 48.0 (47.2–48.8) | 48.1 (47.3–48.9) | 47.4 (46.5–48.3) | 49.0 (48.1–49.8) |
| 45–54 | 50.5 (49.9–51.2) | 48.6 (47.9–49.3) | 49.1 (48.3–49.8) | 47.9 (47.1–48.7) | 48.3 (47.5–49.2) |
| 55–64 | 50.3 (49.7–50.9) | 48.7 (48.1–49.4) | 50.3 (49.6–51.0) | 49.5 (48.8–50.3) | 51.0 (50.3–51.7) |
| ≥ 65 | 51.9 (51.4–52.4) | 51.2 (50.6–51.7) | 53.3 (52.7–53.8) | 52.7 (52.1–53.4) | 53.9 (53.4–54.5) |
| Race | | | | | |
| White | 53.5 (53.2–53.9) | 52.4 (52.1–52.7) | 53.1 (52.8–53.5) | 52.8 (52.4–53.2) | 53.3 (52.9–53.6) |
| Black | 45.2 (44.2–46.2) | 43.6 (42.5–44.6) | 43.6 (42.5–44.7) | 43.2 (42.1–44.3) | 43.8 (42.8–44.9) |
| Other | 50.1 (48.5–51.7) | 47.7 (46.0–49.5) | 51.4 (49.8–53.1) | 49.4 (47.5–51.2) | 50.2 (48.5–51.8) |
| Multiracial | 55.7 (53.1–58.1) | 53.3 (50.5–56.0) | 53.7 (51.4–56.0) | 54.4 (52.2–56.6) | 53.0 (50.9–55.2) |
| Hispanic | 44.6 (43.6–45.6) | 42.5 (41.4–43.5) | 43.7 (42.7–44.7) | 41.7 (40.6–42.8) | 42.5 (41.5–43.6) |
| Employment status | | | | | |
| Employed | 51.2 (50.8–51.6) | 49.7 (49.3–50.2) | 50.6 (50.1–51.0) | 49.9 (49.5–50.4) | 50.9 (50.5–51.4) |
| Unemployed | 51.2 (50.0–52.4) | 49.4 (48.1–50.6) | 48.1 (46.7–49.6) | 46.5 (44.9–48.0) | 46.5 (44.9–48.1) |
| Student | 60.7 (59.0–62.3) | 56.4 (54.7–58.0) | 44.3 (42.6–45.9) | 53.9 (52.1–55.6) | 53.0 (51.3–54.6) |
| Homemaker | 50.6 (49.6–51.7) | 50.2 (49.0–51.5) | 51.8 (50.5–53.1) | 50.6 (49.0–52.1) | 49.9 (48.3–51.4) |
| Retired | 55.4 (54.9–55.9) | 54.6 (54.0–55.2) | 56.3 (55.7–56.8) | 55.8 (55.1–56.5) | 56.6 (55.9–57.1) |
| Unable to work | 30.9 (29.9–32.0) | 28.8 (27.8–29.8) | 30.4 (29.4–31.5) | 30.0 (28.9–31.2) | 26.8 (25.8–27.8) |
| Education | | | | | |
| Did not graduate high school | 38.5 (37.5–39.4) | 36.7 (35.7–37.7) | 38.1 (37.0–39.1) | 36.2 (35.1–37.3) | 35.1 (34.0–36.2) |
| Graduated high school | 46.9 (46.4–47.5) | 45.6 (45.1–46.2) | 46.3 (45.7–46.9) | 45.3 (44.6–45.9) | 45.9 (45.3–46.6) |
| Attended college or technical school | 53.4 (52.8–53.9) | 51.7 (51.1–52.3) | 52.1 (51.5–52.6) | 51.4 (50.8–52.1) | 51.6 (51.0–52.3) |
| Graduated college or technical school | 60.3 (59.8–60.7) | 58.2 (57.8–58.7) | 59.0 (58.6–59.5) | 58.3 (57.8–58.9) | 59.0 (58.5–59.5) |
| Marital status | | | | | |
| Married/partnered | 51.9 (51.5–52.3) | 50.6 (50.2–51.0) | 51.9 (51.5–52.3) | 51.4 (50.9–51.8) | 52.5 (52.0–52.9) |
| Widowed/separated/divorced | 47.0 (46.4–47.5) | 45.0 (44.4–45.6) | 46.3 (45.6–46.9) | 45.4 (44.7–46.1) | 46.6 (45.9–47.2) |
| Single | 52.5 (51.7–53.3) | 50.7 (50.0–51.5) | 50.5 (49.7–51.2) | 49.4 (48.6–50.2) | 47.6 (46.9–48.3) |
| Number of children | | | | | |
| Zero | 51.8 (51.5–52.2) | 50.3 (49.9–50.6) | 51.2 (50.8–51.2) | 50.9 (50.5–51.3) | 50.8 (50.5–51.2) |
| One | 50.0 (49.1–50.8) | 48.6 (47.7–49.5) | 49.4 (48.5–50.3) | 47.4 (46.4–48.4) | 48.6 (47.6–49.5) |
| Two | 50.7 (49.8–51.6) | 48.6 (47.6–49.5) | 49.9 (48.9–50.8) | 48.3 (47.2–49.3) | 49.8 (48.8–50.8) |
| Three or more | 48.0 (46.9–49.2) | 47.1 (45.9–48.3) | 47.3 (46.1–48.5) | 46.3 (45.0–47.6) | 47.1 (45.8–48.4) |
| *Behavioral variables* | | | | | |
| Smoking status | | | | | |
| Current smoker | 45.1 (44.4–45.8) | 43.6 (42.9–44.4) | 44.0 (43.2–44.8) | 43.2 (42.3–44.0) | 44.7 (43.9–45.5) |
| Former smoker | 52.9 (52.4–53.5) | 50.7 (50.2–51.3) | 51.8 (51.2–52.4) | 51.0 (50.4–51.6) | 51.9 (51.3–52.5) |
| Never smoked | 52.4 (52.0–52.8) | 50.8 (50.4–51.2) | 51.6 (51.2–52.1) | 50.9 (50.4–51.4) | 50.7 (50.2–51.1) |
| Fruit and vegetable consumption | | | | | |

*(Continued)*

**Table 2.** (Continued)

| | Sufficient MVPA ($\geq$150 minutes/wk.) | | | | |
|---|---|---|---|---|---|
| | **2011** | **2013** | **2015** | **2017** | **2019** |
| | **% (95%CI)[a]** | **% (95%CI)[a]** | **% (95%CI)[a]** | **% (95%CI)[a]** | **% (95%CI)[a]** |
| Zero | 27.7 (24.0–31.7) | 21.7 (18.6–25.2) | 25.2 (21.6–29.1) | 20.9 (18.1–24.0) | 25.7 (22.7–29.0) |
| One | 39.7 (39.2–40.3) | 38.4 (37.8–39.0) | 39.4 (38.8–40.0) | 37.2 (36.5–37.8) | 39.3 (38.6–39.9) |
| Two | 53.0 (52.5–53.4) | 51.3 (50.9–51.8) | 53.1 (52.6–53.6) | 51.1 (50.6–51.6) | 52.1 (51.6–52.6) |
| Three or more | 62.5 (62.0–63.1) | 61.1 (60.5–61.7) | 60.8 (60.2–61.4) | 59.4 (58.7–60.0) | 59.6 (59.0–60.2) |
| *Health variables* | | | | | |
| Self-Rated Health | | | | | |
| Excellent | 63.1 (62.4–63.8) | 62.3 (61.6–63.0) | 62.1 (61.3–62.8) | 61.6 (60.8–62.4) | 60.9 (60.1–61.7) |
| Very good | 57.3 (56.8–57.8) | 55.6 (55.1–56.1) | 56.3 (55.8–56.9) | 56.5 (55.9–57.1) | 57.3 (56.7–57.9) |
| Good | 47.1 (46.5–47.6) | 45.3 (44.8–45.9) | 47.0 (46.5–47.6) | 46.1 (45.5–46.7) | 47.1 (46.5–47.7) |
| Fair | 37.6 (36.8–38.4) | 35.8 (34.9–36.6) | 36.3 (35.5–37.2) | 35.5 (34.6–36.3) | 36.0 (35.2–36.8) |
| Poor | 25.4 (24.3–26.5) | 24.7 (23.6–25.9) | 25.0 (23.8–26.2) | 25.1 (23.8–26.4) | 24.4 (23.0–25.8) |
| Days of poorer mental health | | | | | |
| Zero | 53.0 (52.7–53.4) | 51.6 (51.2–51.9) | 52.5 (52.2–52.9) | 51.8 (51.4–52.2) | 52.1 (51.7–52.6) |
| One to two | 54.7 (53.7–55.8) | 52.2 (51.1–53.3) | 53.0 (51.9–54.1) | 52.6 (51.4–53.8) | 53.0 (51.9–54.1) |
| Three to six | 51.6 (50.6–52.7) | 49.2 (48.1–50.3) | 49.7 (48.7–50.8) | 49.1 (48.0–50.2) | 50.3 (49.3–51.4) |
| Seven to fourteen | 48.2 (46.8–49.5) | 45.8 (44.4–47.1) | 45.9 (44.5–47.2) | 46.5 (45.1–48.0) | 47.0 (45.7–48.3) |
| Fifteen to thirty | 39.8 (38.9–40.7) | 38.7 (37.8–39.7) | 39.9 (39.0–40.9) | 39.4 (38.4–40.4) | 40.8 (39.9–41.7) |
| Body mass index | | | | | |
| Normal weight | 57.0 (56.4–57.5) | 55.1 (54.5–55.6) | 56.8 (56.2–57.4) | 55.7 (55.0–56.3) | 55.3 (54.7–56.0) |
| Overweight | 53.5 (53.0–54.0) | 51.9 (51.4–52.5) | 52.6 (52.0–53.1) | 52.5 (52.0–53.1) | 53.4 (52.8–54.0) |
| Obesity class I | 46.1 (45.4–46.9) | 45.7 (45.0–46.4) | 46.2 (45.4–46.9) | 45.6 (44.8–46.4) | 47.0 (46.3–47.8) |
| Obesity class II | 40.1 (38.9–41.3) | 38.5 (37.3–39.7) | 40.2 (39.1–41.4) | 40.6 (39.3–41.9) | 41.8 (40.6–43.0) |
| Obesity class III | 33.2 (31.7–34.8) | 32.0 (30.6–33.4) | 33.5 (32.1–34.9) | 32.4 (31.0–33.9) | 33.9 (32.6–35.3) |
| Number of chronic conditions | | | | | |
| Zero | 53.8 (53.2–54.3) | 52.6 (52.1–53.2) | 53.4 (52.8–54.0) | 52.7 (52.1–53.3) | 51.8 (51.2–52.4) |
| One | 54.6 (53.9–55.3) | 52.0 (51.3–52.6) | 53.0 (52.4–53.7) | 52.7 (52.0–53.4) | 52.8 (52.1–53.5) |
| Two | 51.4 (50.7–52.1) | 50.2 (49.5–50.9) | 51.2 (50.5–52.0) | 49.8 (49.0–50.6) | 51.5 (50.8–52.3) |
| Three or more | 43.0 (42.5–43.5) | 42.3 (41.7–42.8) | 42.9 (42.4–43.5) | 42.4 (41.8–43.0) | 44.0 (43.4–44.5) |

[a] Data weighted using stratum weight provided by the Centers for Disease Control and Prevention (CDC).

## Behavioral variables

Compared to current smokers, former smokers were more likely to meet guidelines (APR range = 1.02–1.08). Fruit and vegetable consumption (FVC) were positively correlated with meeting PA guidelines, with each additional serving of fruit or vegetables associated with a slightly increased likelihood of guideline adherence (APR range = 1.01–1.03).

## Health variables

Decreases from 'excellent' SRH resulted in a reduced likelihood of engaging in sufficient PA, with 'poor' SRH being least associated with adherence (APR range = 0.50–0.52). Reporting fifteen to thirty days of poor mental health in a month resulted in the lowest guideline adherence (APRs = $\leq$ 0.96). BMI and guideline adherence were negatively linked, with normal-weight individuals most likely to meet guidelines and obesity class III individuals least likely (APR range = 0.71–0.75). The presence of one or two chronic health conditions was associated with

**Table 3. Adjusted[a] prevalence ratios[b] and 95% confidence intervals for reporting sufficient physical activity for BRFSS samples.**

| | Sufficient MVPA ($\geq$ 150 minutes/week) | | | | |
|---|---|---|---|---|---|
| | **2011** | **2013** | **2015** | **2017** | **2019** |
| | APR[a] (95% CI) | APR[a] (95% CI) | APR[a] (95% CI) | APR[a] (95% CI) | APR[a] (95% CI) |
| *Sociodemographic variables* | | | | | |
| Sex (ref: male) | | | | | |
| Females | 0.93 (0.93–0.93) | 0.93 (0.93–0.93) | 0.95 (0.95–0.95) | 0.94 (0.94–0.94) | 0.92 (0.92–0.92) |
| Age (ref: 18–24 yrs) | | | | | |
| 25–34 | 0.92 (0.92–0.92) | 0.93 (0.93–0.93) | 0.96 (0.96–0.96) | 0.93 (0.93–0.93) | 0.97 (0.97–0.97) |
| 35–44 | 0.93 (0.93–0.93) | 0.97 (0.97–0.97) | 0.96 (0.96–0.96) | 0.96 (0.96–0.96) | 1.03 (1.03–1.03) |
| 45–54 | 0.97 (0.97–0.97) | 1.00 (1.00–1.00) | 1.02 (1.02–1.02) | 0.99 (0.99–0.99) | 1.05 (1.05–1.05) |
| 55–64 | 0.95 (0.95–0.95) | 0.99 (0.98–0.99) | 1.04 (1.04–1.04) | 1.01 (1.01–1.01) | 1.11 (1.11–1.11) |
| 65+ | 0.92 (0.92–0.92) | 0.95 (0.95–0.95) | 1.01 (1.01–1.02) | 0.97 (0.97–0.97) | 1.09 (1.08–1.09) |
| Race (ref: white) | | | | | |
| Black | 0.93 (0.93–0.93) | 0.93 (0.93–0.93) | 0.92 (0.91–0.92) | 0.91 (0.91–0.91) | 0.92 (0.92–0.92) |
| Other | 0.91 (0.90–0.91) | 0.89 (0.89–0.89) | 0.95 (0.95–0.95) | 0.92 (0.92–0.92) | 0.93 (0.93–0.93) |
| Multiracial | 1.10 (1.10–1.10) | 1.09 (1.09–1.09) | 1.08 (1.08–1.08) | 1.11 (1.10–1.11) | 1.07 (1.07–1.08) |
| Hispanic | 0.95 (0.95–0.96) | 0.94 (0.94–0.94) | 0.95 (0.95–0.95) | 0.92 (0.92–0.92) | 0.94 (0.94–0.94) |
| Employment status (ref: employed) | | | | | |
| Unemployed | 1.14 (1.14–1.14) | 1.13 (1.13–1.13) | 1.08 (1.08–1.08) | 1.08 (1.08–1.09) | 1.05 (1.05–1.05) |
| Homemaker | 1.12 (1.12–1.12) | 1.16 (1.16–1.16) | 1.15 (1.15–1.15) | 1.16 (1.16–1.16) | 1.15 (1.15–1.15) |
| Student | 1.12 (1.12–1.12) | 1.09 (1.09–1.09) | 1.09 (1.08–1.09) | 1.05 (1.05–1.05) | 1.08 (1.08–1.08) |
| Retired | 1.20 (1.20–1.20) | 1.21 (1.21–1.21) | 1.19 (1.19–1.19) | 1.20 (1.20–1.20) | 1.14 (1.14–1.14) |
| Unable to work | 0.93 (0.93–0.93) | 0.89 (0.88–0.89) | 0.89 (0.89–0.90) | 0.91 (0.91–0.91) | 0.76 (0.76–0.76) |
| Education level (ref: did not graduate high school) | | | | | |
| Graduated high school | 1.09 (1.09–1.09) | 1.10 (1.10–1.10) | 1.07 (1.07–1.07) | 1.10 (1.10–1.10) | 1.15 (1.15–1.15) |
| Attended college or technical school | 1.19 (1.19–1.19) | 1.21 (1.20–1.21) | 1.16 (1.16–1.16) | 1.20 (1.20–1.21) | 1.23 (1.23–1.23) |
| Graduated college or technical school | 1.28 (1.28–1.28) | 1.27 (1.27–1.27) | 1.24 (1.24–1.24) | 1.28 (1.28–1.28) | 1.32 (1.32–1.32) |
| Marital status (ref: single) | | | | | |
| Married/partnered | 0.98 (0.98–0.98) | 0.97 (0.97–0.97) | 0.97 (0.96–0.97) | 1.00 (1.00–1.00) | 1.02 (1.01–1.02) |
| Divorced/widowed/separated | 0.98 (0.98–0.98) | 0.95 (0.95–0.95) | 0.95 (0.95–0.95) | 0.97 (0.97–0.97) | 1.00 (1.00–1.00) |
| Number of children (ref: zero) | | | | | |
| One | 0.99 (0.99–0.99) | 1.00 (1.00–1.00) | 1.00 (1.00–1.00) | 0.98 (0.97–0.98) | 1.01 (1.01–1.01) |
| Two | 0.99 (0.99–0.99) | 0.98 (0.98–0.98) | 1.00 (1.00–1.00) | 0.96 (0.96–0.96) | 1.01 (1.01–1.01) |
| Three or more | 0.99 (0.99–0.99) | 1.00 (1.00–1.00) | 1.00 (1.00–1.00) | 0.97 (0.97–0.97) | 1.00 (1.00–1.00) |
| *Behavioral Variables* | | | | | |
| Smoking status (ref: current smokers) | | | | | |
| Former smokers | 1.08 (1.08–1.08) | 1.06 (1.06–1.06) | 1.05 (1.05–1.06) | 1.05 (1.05–1.05) | 1.02 (1.01–1.02) |
| Never smoked | 1.04 (1.04–1.04) | 1.03 (1.03–1.03) | 1.02 (1.02–1.02) | 1.02 (1.02–1.02) | 0.99 (0.98–0.99) |
| Fruit and vegetable consumption | | | | | |
| (Change with +1 serve) | 1.03 (1.03–1.03) | 1.03 (1.03–1.03) | 1.03 (1.03–1.03) | 1.01 (1.01–1.01) | 1.02 (1.02–1.02) |
| *Health variables* | | | | | |
| Self-rated health (ref: excellent) | | | | | |
| Very good | 0.92 (0.92–0.92) | 0.92 (0.92–0.92) | 0.92 (0.92–0.92) | 0.93 (0.93–0.93) | 0.94 (0.94–0.94) |
| Good | 0.82 (0.82–0.82) | 0.81 (0.81–0.81) | 0.83 (0.83–0.83) | 0.82 (0.82–0.82) | 0.83 (0.83–0.83) |
| Fair | 0.70 (0.70–0.70) | 0.68 (0.68–0.68) | 0.69 (0.69–0.69) | 0.68 (0.68–0.68) | 0.69 (0.69–0.69) |
| Poor | 0.51 (0.51–0.51) | 0.51 (0.51–0.51) | 0.50 (0.50–0.50) | 0.52 (0.52–0.52) | 0.51 (0.51–0.51) |
| Number of poor mental health days (ref: zero) | | | | | |
| One to two | 1.03 (1.03–1.03) | 1.01 (1.01–1.01) | 1.00 (1.00–1.00) | 1.02 (1.02–1.02) | 1.01 (1.01–1.01) |

*(Continued)*

**Table 3.** (Continued)

| | Sufficient MVPA ($\geq$ 150 minutes/week) | | | | |
|---|---|---|---|---|---|
| | **2011** | **2013** | **2015** | **2017** | **2019** |
| | APR[a] (95% CI) | APR[a] (95% CI) | APR[a] (95% CI) | APR[a] (95% CI) | APR[a] (95% CI) |
| Three to six | 1.01 (1.01–1.01) | 1.00 (1.00–1.00) | 0.98 (0.98–0.98) | 1.00 (1.00–1.00) | 1.01 (1.01–1.01) |
| Seven to fourteen | 1.00 (1.00–1.00) | 0.98 (0.98–0.98) | 0.97 (0.97–0.97) | 1.00 (1.00–1.00) | 1.00 (1.00–1.00) |
| Fifteen to thirty | 0.93 (0.93–0.93) | 0.95 (0.95–0.95) | 0.96 (0.96–0.96) | 0.95 (0.95–0.95) | 0.96 (0.96–0.97) |
| Body mass index (ref: normal weight) | | | | | |
| Overweight | 0.98 (0.98–0.98) | 0.97 (0.97–0.97) | 0.96 (0.96–0.96) | 0.97 (0.96–0.97) | 0.97 (0.97–0.97) |
| Obesity class I | 0.90 (0.90–0.90) | 0.91 (0.91–0.91) | 0.89 (0.89–0.89) | 0.89 (0.89–0.89) | 0.91 (0.91–0.91) |
| Obesity class II | 0.81 (0.81–0.81) | 0.81 (0.81–0.81) | 0.82 (0.82–0.82) | 0.84 (0.84–0.84) | 0.85 (0.85–0.85) |
| Obesity class III | 0.73 (0.73–0.73) | 0.72 (0.72–0.72) | 0.71 (0.71–0.72) | 0.72 (0.72–0.72) | 0.75 (0.75–0.75) |
| Number of chronic health conditions (ref: zero) | | | | | |
| One | 1.04 (1.04–1.04) | 1.02 (1.02–1.02) | 1.01 (1.01–1.01) | 1.02 (1.02–1.02) | 1.03 (1.03–1.03) |
| Two | 1.03 (1.03–1.03) | 1.02 (1.02–1.02) | 1.02 (1.02–1.02) | 0.99 (0.99–0.99) | 1.03 (1.03–1.03) |
| Three or more | 0.99 (0.99–0.99) | 0.99 (0.99–0.99) | 0.96 (0.96–0.96) | 0.95 (0.95–0.95) | 0.99 (0.99–0.99) |

[a] Adjusted for all explanatory variables in the model

[b] Prevalence Ratios calculated by Poisson log-linear regression with a robust-error variance

a slightly increased likelihood of meeting guidelines (APR range = 1.00–1.04), whereas having three or more conditions was associated with a slightly decreased likelihood of meeting guidelines (APR range = 0.95–1.00).

## Discussion

This study described the temporal trends of adults adhering to current aerobic PA guidelines and identified significant sociodemographic, behavioral, and health-related correlates for guideline adherence in representative samples of U.S. adults. The key findings were that approximately half of each sample reported sufficient PA across each survey and guideline adherence among young adults (18–24 years) declined over time. Noteworthy correlates of adhering to guidelines included male sex, identifying as white, higher levels of education, former and never smokers, greater fruit and vegetable consumption and normal body mass index.

Previous self-reported prevalence estimates of MVPA guideline adherence in U.S adults ranged from 50.5% to 65.2% [13, 14]. Adherence to MVPA guidelines among BRFSS respondents between 2011 and 2019 was lower than expected, as approximately half of each sample (range 49.5–51.1%) reported 150 or more minutes of MVPA per week during the month before their interview in each of the surveys. This suggests the true prevalence of sufficient aerobic activity may be in the lower range of current population estimates [13, 14] and closer to those reported in this study.

### Sociodemographic variables

One sociodemographic correlate of MVPA guideline adherence was being male, with the contrast between the sexes in this study corroborating others that have shown that men are generally more physically active than women [11, 15, 34, 35]. While the differences between males and females in this study are significant, previous literature has suggested that the variation in adherence to PA guidelines is due to many factors. Some evidence has indicated that varying types, frequencies and intensities of PA are a driver behind the discrepancy between guideline

adherence between males and females [36]. Other evidence has suggested that interpersonal and social-cultural factors within the social-ecological model for health are notable contributors to reduced PA participation for females that are not commonly experienced by males [37]. Gender norms often reinforce expectations that prioritize female caregiving duties associated with mother/wife identity, causing PA participation to result in feelings of guilt as it is not viewed as a 'feminine' activity [37]. Societal norms such as perpetuated stereotypes and lack of a female-specific PA culture often discourage engagement in activities due to the perception that PA will be viewed as masculine or result in unwanted judgement from others. Beauty and media standards are considerable factors that act as a barrier to engaging in PA, as it is viewed that various activities will promote unwanted muscle development which will be counterproductive to the commonly promoted slender physique [37].

Differences between guideline adherence were observed among racial/ethnicity groups, with white individuals more likely to meet MVPA guidelines compared to other groups, aside from multiracial individuals. For example, previous studies have reported that individuals identifying as white were more likely to engage in non-occupational PA compared to other racial groups [15, 38, 39], with limited contrasting evidence [19, 40]. Other sociodemographic correlates of MVPA guideline adherence were level of education and employment status, as those who attended or graduated college or technical school and were not employed were more likely to report adequate MVPA. The evidence pertaining to education level and PA behaviors is consistent and has indicated higher levels of education are associated with increased PA [15, 41, 42]. The current study aligns with evidence suggesting that employed individuals are at risk of being inactive [43], although other evidence has suggested that the two variables are positively related, with those employed more likely to be active [15]. A potential explanation for this finding is that employed individuals have less free time to engage in PA compared to the other employment subgroups on account of their employment.

The relationship between age and MVPA guideline adherence in this study was unexpected, as evidence has largely indicated age is inversely related to engagement in PA [15, 17, 18], or that no association exists between the variables [19]. While the APRs for sufficient PA were higher for individuals aged 55 or older in 2019, this can likely be attributed to the decline in PA among the 18-to-24-year-old reference group. This decline in sufficient PA among young adults was unexpected, particularly as prior evidence has suggested that increasing age is generally associated with declines in PA [44]. A potential explanation for this decline is the increased accessibility and reliance on sedentary behaviors associated with technological advancements, resulting in decreased PA and increased sedentary behaviors among young adults. An analysis of high school students in the United States found that TV screen time significantly decreased and other screen time significantly increased between 2007 and 2015 [45]. The authors concluded that an increasing proportion of young people were not engaging in sufficient PA due to excessive screen time. It is therefore plausible that these behaviors continued into young adulthood for a large proportion of United States adults. However, the reduced likelihood of guideline adherence for respondents aged between 25 and 44 is supportive of some previous research investigating the impact of major life events and transitions on PA behaviors. Major life events such as marriage, pregnancies and increasing parental responsibilities, as well as career development, are very common during this life stage and have been attributed to reduced PA behaviors [46]. The associations with PA previously reported between relationship [46–48] and parental status [46, 48, 49] were observed in the current study but to a lesser extent. Married/partnered and divorced/widowed/separated were less likely to adhere to guidelines compared to single individuals (APR range = 0.95–1.00), aside from 2019, while the presence of one or more children in a given household was associated with slightly lower guideline adherence only in 2017 (APR range = 0.96–0.98).

## Behavioral variables

The findings from this study support previous associations identified among former smokers and individuals who had never smoked, as they were more likely to adhere to MVPA guidelines when compared to current smokers. Previous literature about smoking and PA behaviors has shown the two behaviors are incongruent [50, 51], with limited contrasting evidence [16]. Prior investigations into the relationship between fruit and vegetable consumption (FVC) and PA have provided conflicting results. While some studies have concluded that an association between FVC and engaging in PA exists [52–55], others have suggested the behaviors are unrelated [54, 56]. Though a temporal link could not be explored between the variables in this study, the results suggest that FVC is only modestly correlated with sufficient PA as each additional serving from zero was associated with a 1–3% increased likelihood of meeting guidelines (APR range = 1.01–1.03).

## Health variables

The protective effects of PA on the risk of poorer mental health symptoms [57, 58] and mental health disorders [4, 59] have already been well-established, among individuals with and without relevant disorders. In the current study, although very minimal differences were observed between most groups, a reduced likelihood of adherence was experienced for individuals with fifteen or more days of poorer mental health days. The relationship between SRH and PA in this study also showed an inverse association, confirming results from previous studies. In these studies, an association between SRH and PA were present in a dose-response manner [60–62]. A study of older adults [62] found that PA was 51% higher among individuals with very good SRH compared to those with poor or very poor SRH.

Current evidence has shown that chronic disease status [4] and obesity [15] are notable correlates of insufficient activity. This study provides mixed results about chronic disease status and being sufficiently active. Individuals who reported one or two conditions had a slightly higher prevalence of adhering to guidelines compared to the disease-free reference group. While it is plausible that the association between chronic disease and PA observed for individuals with one or two conditions could be explained by increases in activity in individuals who were previously insufficiently active after receiving advice from a health professional, this is unlikely. A study of Australian women concluded that being diagnosed with a chronic disease did not affect PA behaviors [63].

The association between disease status and guideline adherence in this study for individuals with three or more conditions were expected, as prior evidence has indicated that guideline adherence is low for individuals with multimorbidity [64]. The relationship between obesity and PA behaviors has been heavily investigated, with a systematic review [65] of obese adults highlighting significant associations with physical inactivity. Studies comparing PA behaviors across BMI classes have yielded similar results, with PA behaviors decreasing with increasing levels of adiposity [13, 66–68]. The findings from the current study align with the aforementioned evidence, reinforcing the importance of a healthy body weight and prevention of unhealthy weight gain.

## Strengths and limitations

Strengths of this study include the large, nationally representative sample of U.S. adults, standardized processes of recruitment and data collection that allow comparison of results with further BRFSS studies and similar explorations. Another strength was the inclusion of a weighting variable that helped to correct for non-response, stratification, and clustering, improving the generalizability of the findings to the total U.S. adult population.

While interpreting these results, some limitations need to be considered. Firstly, the reliance on self-report measures of MVPA is prone to recall bias and may not encapsulate habitual PA behaviors [25]. The limiting cross-sectional nature of this study also does not allow for causal exploration between the various exposure variables and guideline adherence. Future studies exploring correlates and determinants of PA that include objective assessments of MVPA would help to verify temporal associations between PA behaviors with currently ambiguous factors. Additionally, the potential influence of seasonality on PA behaviors was not accounted for in the analysis. The seasonality distribution of BRFSS interviews was relatively consistent across all surveys included in the study, suggesting that seasonal variations are likely to average out in such a large sample. For the seasonality distribution of BRFSS survey interviews refer to S1 Table.

## Conclusion

The findings from this large study suggest that the prevalence of adherence to MVPA guidelines in the adult United States population falls within the lower range of previous estimates, with approximately half of the population reporting sufficient aerobic PA. Many previously identified correlates of sufficient activity were also observed in this study. Notable findings from this study were that employed individuals were noticeably less likely to adhere to MVPA guidelines, while modest increases in likelihood were associated with a single-serving increase of fruit and vegetables. Sustained levels of population inactivity identified in this study will continue to burden healthcare systems due to the identified associations with morbidity and mortality, further emphasizing the importance of continued public health approaches to tackle physical inactivity. The identified associations between sociodemographic, behavioral and health-related variables and guideline adherence in this study emphasize the complexity surrounding tackling physical inactivity. Innovative approaches promoting PA tailored to both general populations and at-risk populations are imperative to help reduce the incidence of non-communicable diseases and contribute to advancements in achieving the WHO's goal for reducing inactivity by 15% by 2030.

## Supporting information

**S1 Table. Seasonality distribution of conducted BRFSS interviews for 2011, 2013, 2015, 2017 and 2019 surveys.**
(DOCX)

## Acknowledgments

All authors are grateful to the respondents of the 2011, 2013, 2015, 2017 and 2019 Behavioral Risk Factor Surveillance System for their time in participating in this study.

## Author Contributions

**Conceptualization:** David Abernethy, Jason Bennie, Toby Pavey.

**Data curation:** David Abernethy.

**Formal analysis:** David Abernethy, Toby Pavey.

**Investigation:** David Abernethy.

**Methodology:** David Abernethy, Jason Bennie, Toby Pavey.

**Project administration:** David Abernethy, Toby Pavey.

**Supervision:** Jason Bennie, Toby Pavey.

**Writing – original draft:** David Abernethy.

**Writing – review & editing:** David Abernethy, Jason Bennie, Toby Pavey.

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
