## [Decision Letter · Decision Letter 0]

20 Sep 2024

PONE-D-24-25566Temporal trends in aerobic physical activity guideline adherence among nationally representative samples of U.S adults between 2011 and 2019: cross-sectional findings from a sample of over 2 million adultsPLOS ONE

Dear Dr. Abernethy,

Thank you for submitting your manuscript to PLOS ONE. After careful consideration, we feel that it has merit but does not fully meet PLOS ONE’s publication criteria as it currently stands. Therefore, we invite you to submit a revised version of the manuscript that addresses the points raised during the review process.

We look forward to receiving your revised manuscript.

Kind regards,

Sandhya Yadav

Academic Editor

PLOS ONE

Journal Requirements:

Reviewers' comments:

Reviewer's Responses to Questions

**Comments to the Author**

1. Is the manuscript technically sound, and do the data support the conclusions?

Reviewer #1: Yes

2. Has the statistical analysis been performed appropriately and rigorously? 

Reviewer #1: Yes

3. Have the authors made all data underlying the findings in their manuscript fully available?

Reviewer #1: Yes

4. Is the manuscript presented in an intelligible fashion and written in standard English?

Reviewer #1: Yes

5. Review Comments to the Author

Reviewer #1: The authors analysed a large set of data for many years including a large number of participants which makes the results more robust and comprehensive.

However, I was wondering why the seasonality effect was not among the factors / variables assessed by the study. Were all the surveys conducted at the same period each year or not? As the main question was to answer based on what happened last month?

The point here is, if these surveys were conducted in different periods of the year, i.e. summer, winter, fall, etc. then there might be a bias that need to be adjusted for, at least statistically. Presenting a table indicating when exactly (month/season) the surveys were conducted each year would give an insight of this aspect.

Finally, considering this aspect as a one of the weakness of the study in the limitation section would be important.

6. PLOS authors have the option to publish the peer review history of their article (what does this mean?). If published, this will include your full peer review and any attached files.

Reviewer #1: **Yes: **Mahamat Fayiz Abakar

---

## [Author Response · Author response to Decision Letter 0]

9 Oct 2024

Below is the same information included in the response letter attached to this form:

Response Letter for PLOS ONE Manuscript [PONE-D-24-25566]

Dear Dr Mahamat Fayiz Abakar,

We appreciate the time and effort that you invested in providing valuable feedback on our manuscript, "Temporal Trends in Aerobic Physical Activity Guideline Adherence Among Nationally Representative Samples of U.S Adults Between 2011 and 2019: Cross-sectional Findings from a Sample of Over 2 Million Adults" [PONE-D-24-25566]. We have carefully considered your suggestions and have responded to each point in detail below.

Reviewer Comments: 

"I was wondering why the seasonality effect was not among the factors / variables assessed by the study. Were all the surveys conducted at the same period each year or not? As the main question was to answer based on what happened last month?

The point here is, if these surveys were conducted in different periods of the year, i.e. summer, winter, fall, etc. then there might be a bias that need to be adjusted for, at least statistically. Presenting a table indicating when exactly (month/season) the surveys were conducted each year would give an insight of this aspect.

Finally, considering this aspect as a one of the weakness of the study in the limitation section would be important.”

We appreciate your insightful comment regarding the potential influence of seasonality on physical activity behaviors. We acknowledge that seasonality can impact physical activity, as environmental factors like temperature and daylight availability may influence individuals’ capacity or motivation to engage in outdoor physical activity. 

However, seasonality has not historically been accounted for in studies using BRFSS data. The timing of the surveys is spread across the year, likely resulting in seasonal fluctuations averaging out over such a large sample. Furthermore, adjusting for seasonality in our study focused on long term trends could shift the focus towards more cyclical fluctuations that are not relevant to our study’s primary research questions and potentially dilute the interpretations of long-term population activity adherence. 

To address your concern, we have added a supplementary table (S1 Table in the revised manuscript) outlining the seasonal distribution of when the BRFSS survey interviews were conducted to provide readers with some insight into this aspect. The distribution of the survey interviews was relatively similar across the seasons for all BRFSS surveys used in our study. Additionally, we have added a statement to the limitations section acknowledging that seasonality may influence activity levels of individuals and to take this into account when interpreting the study’s results.

We sincerely thank you again for your careful consideration and valuable feedback. We hope that the revisions we have made address your concerns adequately. We believe that these changes further enhance the rigor and clarity of our manuscript, and we look forward to your final evaluation. 

Sincerely, 

David Abernethy

---

## [Decision Letter · Decision Letter 1]

5 Dec 2024

Temporal trends in aerobic physical activity guideline adherence among nationally representative samples of U.S adults between 2011 and 2019: cross-sectional findings from a sample of over 2 million adults

PONE-D-24-25566R1

Dear Dr. Abernethy,

We’re pleased to inform you that your manuscript has been judged scientifically suitable for publication and will be formally accepted for publication once it meets all outstanding technical requirements.

Kind regards,

Henri Tilga, PhD

Academic Editor

PLOS ONE

Additional Editor Comments (optional):

Reviewers' comments:

Reviewer's Responses to Questions

**Comments to the Author**

1. If the authors have adequately addressed your comments raised in a previous round of review and you feel that this manuscript is now acceptable for publication, you may indicate that here to bypass the “Comments to the Author” section, enter your conflict of interest statement in the “Confidential to Editor” section, and submit your "Accept" recommendation.

Reviewer #1: All comments have been addressed

2. Is the manuscript technically sound, and do the data support the conclusions?

Reviewer #1: Yes

3. Has the statistical analysis been performed appropriately and rigorously? 

Reviewer #1: Yes

4. Have the authors made all data underlying the findings in their manuscript fully available?

Reviewer #1: Yes

5. Is the manuscript presented in an intelligible fashion and written in standard English?

Reviewer #1: Yes

6. Review Comments to the Author

Reviewer #1: Dear authors,

Thank you for your revised manuscript in which my comments have been taken into consideration. Your paper will contribute to highlight the issues related to physical activity, a domain that remains largely under covered.

7. PLOS authors have the option to publish the peer review history of their article (what does this mean?). If published, this will include your full peer review and any attached files.

Reviewer #1: **Yes: **Mahamat Fayiz Abakar

---

## [Editor Report · Acceptance letter]

10 Dec 2024

PONE-D-24-25566R1 

PLOS ONE

Dear Dr. Abernethy, 

I'm pleased to inform you that your manuscript has been deemed suitable for publication in PLOS ONE. Congratulations! Your manuscript is now being handed over to our production team.

Kind regards, 

on behalf of

Dr. Henri Tilga 

Academic Editor

PLOS ONE